# The Use of DAO as a Marker for Histamine Intolerance: Measurements and Determinants in a Large Random Population-Based Survey

**DOI:** 10.3390/nu15132887

**Published:** 2023-06-26

**Authors:** Jenny van Odijk, Adina Weisheit, Monica Arvidsson, Nicolae Miron, Bright Nwaru, Linda Ekerljung

**Affiliations:** 1Department of Respiratory Medicine and Allergology, Sahlgrenska University Hospital, 413 46 Gothenburg, Sweden; adina.weisheit@vgregion.se (A.W.);; 2Department of Internal Medicine and Clinical Nutrition, Institute of Medicine, Sahlgrenska Academy, University of Gothenburg, 405 30 Gothenburg, Sweden; 3Department of Clinical Immunology, Sahlgrenska University Hospital, 413 46 Gothenburg, Sweden; 4Krefting Research Centre, Institute of Medicine, University of Gothenburg, 405 30 Gothenburg, Sweden

**Keywords:** histamine intolerance, diamine oxidase, food intolerance, IgE-mediated food allergy

## Abstract

Histamine intolerance (HIT) is a common adverse reaction to food where elimination and reintroduction of histamine-rich food is part of the investigation. Analysis of the enzyme diamine oxidase (DAO) is sometimes used as an additional tool for diagnosis. This study aimed to describe the distribution of DAO in a large representative cohort of adults and to determine the association between DAO activity and possible associated factors. The study is based on the population-based West Sweden Asthma Study and includes 1051 subjects. Subjects underwent structured interviews including questions on demography, asthma, allergy symptoms, and lifestyle factors. Subjects were assessed for specific-IgE-antibodies and measurement of DAO activity in serum. Previously suggested cut-off levels for low values (<3 U/mL), normal values (>10 U/mL), and median levels of DAO were used. In the group of 1051 subjects, only a few presented reactions upon histamine intake, whereas 44% presented DAO levels below the suggested normal cut-off levels. BMI and age were shown to have an impact on DAO activity among women with increasing activity of DAO with increasing BMI and age. Among men, only increasing age was seen to have an impact on DAO levels. There was no difference in DAO levels with different sensitization status to common foods or airborne allergens. No association between DAO levels and reported symptoms to histamine-rich foods could be found. In conclusion, the determination of the DAO enzyme needs to be re-evaluated and may not be used as a valuable tool for histamine intolerance using current cut-off values. Further studies are needed to improve the use of DAO as a biomarker for histamine intolerance.

## 1. Introduction

According to World Allergy Organization, histamine intolerance (HIT) is classified as a non-allergic food hypersensitivity [1,2] and can be defined as a disorder where reduced histamine degradation capacity in the intestines leads to the appearance of adverse symptoms possibly caused by impaired diamine oxidase (DAO) activity [3,4]. The prevalence of this condition is unknown because estimates of prevalence of different intolerances vary greatly, depending on many factors such as study population and classification. About one out of five persons suffers from some kind of food intolerance where reactions to ingested histamine can be included [2]. 

The investigation of HIT is based on the relevant anamnesis for food hypersensitivity (elimination and provocation) where combination of diagnostic criteria includes display of typical clinical symptoms and exclusion of other related disorders [2,5]. The most frequently described symptoms upon dietary histamine intake are heterogeneous and differential diagnosis must be considered before the setting of the diagnosis [6]. Typical symptoms are related to dermatological manifestations, including eczema, flushing, and urticaria, as well as gastrointestinal symptoms such as abdominal pain and alterations in bowel movements. Symptoms associated with the cardiovascular system, such as hypotension and tachycardia, have also been described [4,7].

Histamine is a biogenic amine deriving from the essential amino acid histidine [8]. Histamine can be endogenously synthesized and stored in mast cells and in basophils. It is an important mediator involved in the IgE-mediated allergic reaction. Histamine, both endogenously synthesized and ingested through diet, can be metabolized through two pathways, where Methylation by histamine-N-methyltransferase (HNMT) and degradation by diamine oxidase (DAO) are enzymes involved in the process. DAO has been considered as responsible for the degradation of histamine-containing foods [5].

Histamine and other biogenic amines in foods and drinks are produced through bacterial decarboxylation and are present to various degrees in a wide range of products [9]. High concentrations are, for example, found in matured cheeses, processed meat, processed fish, and fermented drinks, such as beer and wine [8].

Beyond clinical investigations, manifestations of histamine intolerance may be related to levels or activity of DAO enzyme [10]. In what way the reference values for determination of DAO can be connected to intolerance needs to be further elucidated. Based on earlier studies, the normal concentration range of DAO below 3 U/mL have been connected to higher risk of histamine intolerance and is improbable in subjects with >10 U/mL [11].

In a prospective study of subjects with clinically suspected histamine intolerance, DAO levels were significantly lower compared to levels in control subjects [12], but the reliability of measurement of DAO activity in serum has also been questioned. In one case control study, no correlation between serum levels of DAO and clinical status of histamine intolerance was found [13]. A recently published cohort study with a retrospective design found low levels of DAO to be a reliable biomarker for histamine intolerance [10]. A few studies further addressing this question have been published locally in non-English journals, complicating the interpretation of the results and the research area.

While interest in histamine intolerance has considerably grown in recent years, data on measurements of DAO enzyme in representative cohorts are still lacking. Thus, while DAO is thought to be a tool in diagnosing histamine intolerance, there are still considerable uncertainties regarding its diagnostic value on a population level. More scientific evidence is still required to define, diagnose, and clinically manage histamine intolerance. There are still some basic questions that need to be answered and elucidated upon, including how DAO is distributed among a larger population including subjects with and without certain comorbidities.

Therefore, the aim of this study is to describe the distribution of DAO enzyme in a large representative cohort of adults and to determine levels of DAO in relation to factors including age, sex, BMI, atopy, and described food intolerance.

## 2. Materials and Methods

This is a cohort study based on the West Sweden Asthma Study (WSAS). WSAS is a population-based longitudinal study established in West Sweden in 2008, which has been described in detail elsewhere [14]. Briefly, in 2008, a questionnaire survey was sent to 30,000 randomly selected adults aged 16–75 years in western Sweden, of which 18,087 (67%) responded. A non-response study performed showed no differences in prevalence of symptoms or disease between responders and non-responders. Out of the responders, a random sample of 2000 was invited to additional clinical examinations. In total, 1172 (59%) participated. The participants underwent a thorough structured interview that included questions on demography, asthma, and allergy symptoms, as well as lifestyle factors. They also answered several self-administered questionnaires, one of which included questions on food hypersensitivity, as well as other hypersensitivity symptoms described in detail elsewhere [15]. The participants were clinically examined with anthropometric measurements and. their lung function was assessed with spirometry. Additionally, a skin prick test was performed to assess allergy to airborne allergens and blood was donated for, among other things, specific IgE testing, DAO measurement, and blood count. The sample included in the current survey were the subjects for which DAO-measurements was available (1051 subjects). In analyses where DAO and its relation to food hypersensitivity were included, the sample size was 936 subjects. 

### 2.1. Ethical Considerations

The study was approved by the regional ethics board in Gothenburg, no. 593-08. All participation was voluntary, and all participants signed a written consent form with possibility to withdraw their data at any time. In the clinical examinations, if something required a medical intervention or follow up, participants was referred to their local general practitioner. 

### 2.2. Questionnaire 

In total, 56 food items for which data for self-reported symptoms were available, indicating food hypersensitivity. For each food item, the participants described associated symptoms or if they practiced total avoidance. Symptoms were grouped into the following areas: gastro-intestinal, skin, airway, and neurological. 

Of the 56 food items, there were 10 for which specific IgE was measured: egg white, milk, fish, wheat, soybean, almond, peanut, hazelnut, Brazil nut, and coconut. According to earlier publications and analysis, 9 food items were considered as histamine containing foods and were used for the analysis of reaction to histamine in foods [8]: tomato, chocolate, cheese, fish, shellfish, cured meats, beef, pork, and beer/wine. 

### 2.3. Analysis of DAO 

The levels of DAO activity in human serum samples were measured with a commercially available kit (DAO-REA Sciotec, HS 421-37 Tulln an der Donau, Austria) by incubating serum with radiolabeled putrescine-dihydrochloride as a substrate. DAO degrades putrescine until Δ1 pyrroline that contained the initial radiolabel. After an additional step, which ensured that Δ1 pyrroline was extracted from the reaction medium, the amount of radioactivity was measured with a beta-counter (Perkin-Elmer MicroBeta 2). The levels of radioactive Δ1 pyrroline were directly proportional with the activity of DAO. According to manufacturer levels, ≥10 U/mL was considered as normal levels of DAO. Moreover, 3–10 U/mL were considered to be slightly decreased levels and <3 U/mL were considered to be distinctively decreased levels of DAO. 

### 2.4. Analysis of IgE Antibodies 

Blood samples were collected to assess the specific IgE-sensitization profile of 2 ImmunoCAP allergen panels: fx1 (peanut, hazel nut, Brazil nut, almond, coconut) and fx5 (egg white, milk, fish, wheat, peanut, soybean) (Thermo Fischer Scientific, Uppsala, Sweden). All positive samples to a panel (with a value of >0.35 kUA/L) were further assessed for the individual allergens included in that panel, according to the manufacturer’s instructions. 

### 2.5. Statistical Analyses

Statistical analyses were performed using SPSS version 28 (IBM Corp, New York, NY, USA). DAO was non-normally distributed. In this case, medians with interquartile ranges (IQR) were presented. For normally distributed variables, means with standard deviations (SD) were presented. To compare normally distributed variables, the Student’s *t*-test was used. To compare distributions of non-normally distributed continuous data, the Mann–Whitney U-test was used. Categorical variables were compared using Chi-squared tests. Moreover, *p*-values were calculated with the Fisher’s exact test for two categories: Pearson’s chi-squared test for >2 nominal categories and the Mantel–Haenszel test for trend for >2 ordinal categories. A *p*-value of <0.05 was considered statistically significant throughout the study. Correlation coefficients (r) were used to explore associations between DAO-level and sex, age, and BMI, respectively. Spearman’s r was used to best fit the data. To follow up on the correlation analysis, multinominal logistic regression was used to analyze factors associated with DAO. The results are presented as odd ratios with 95% CI with DAO level divided in three categories. Covariates included were sex, age, and BMI.

## 3. Results

### 3.1. Demographic Data

In the cohort where DAO was analyzed (N 1051), 12% had asthma and 27% were sensitized to any airborne allergen. Almost 5.5% were sensitized to common food allergens (egg white, milk, fish, wheat, peanut, and soybean). About one out of three declared the use of antihistamines or other medication for hay fever. The subjects available for DAO testing did not differ significantly from the group where DAO analysis was not available, regarding age, sex, BMI, atopic conditions, or any other specified condition (Table 1).

### 3.2. Distribution of DAO Activity 

The median value for DAO level was 11.1 U/mL, with an IQR of 11.3 U/mL. DAO activity measured in the serum was widely distributed within the group with a positively skewed distribution (Figure 1). When considering the suggested cut-off levels, a DAO activity level less than 10 U /mL was found in 44%; distinctively decreased level of DAO (below 3 U/mL) was found in less than 5% of the cohort. Hence, only 56% of the population was within what is considered the normal range of DAO for individuals without histamine intolerance.

### 3.3. Factors Associated with DAO Levels

When considering the median DAO, a significant trend of increasing DAO was seen with increasing BMI (*p* = 0.004) and increasing age (*p* <0.001), where the group >61 years were found to have a higher DAO than other age groups (Figure 2). No sex differences in median DAO were observed. The increase in the median DAO with increasing BMI was only present among women, while the correlation with increasing age was seen in both sexes (Figure 3). 

When applying suggested cut-off levels for DAO, distinctively low levels below 3 U/mL were less common in older subjects (<61 years of age) than in younger subjects (*p* < 0.001). Furthermore, low DAO levels less than 3 U/mL were also more common in subjects with BMI <20 than in those with higher BMI (*p* < 0.03) (Table 2). A tendency of low levels being more common among women than in men was also seen. DAO activity among subjects with or without asthma, atopic conditions, or other medical conditions did not differ with stated cut-off levels, but DAO activity less than 3 U/mL (*p* < 0.05) was more common among users of medication for hay fever than non-users. 

In the multinominal logistic regression where sex, age, and BMI were taken into the model, only age remained as a risk factor for having a DAO below 3 U/mL, with the highest odds ratio for those ages being 61–76 years (OR 4.30, 95%CI 1.57–11.82) when compared to DAO ≥ 10 (Table 3). The protective effect of increasing BMI for having a DAO <3 was no longer significant when the other covariates were included in the model. None of the included covariates remained a significant risk factor for the intermediate group with DAO 3–10 U/mL when compared to DAO ≥10 (Table 3).

### 3.4. Symptoms Association with Foods Rich in Histamine

Symptoms often related to histamine hypersensitivity (gastrointestinal symptoms, any skin symptoms, any airway, or any headache/dizziness) was reported by 8.8% of the total group. No difference in median DAO could be seen between subjects reporting symptoms connected to the condition or no reported symptoms (Table 4). Among subjects reporting symptoms to histamine rich foods, the most common symptoms were gastrointestinal (5.9%) and airway (2.1%) symptoms. Airway symptoms were significantly more common among subjects with DAO < 3 U/mL (8.3%), especially when compared with DAO ≥10 U/mL (1.9%, *p* = 0.03).

## 4. Discussion

This is the first study to describe the distribution of DAO in a population-based cohort with no known history of histamine intolerance. In the current study of more than 1000 adults, a considerable number of subjects reported not having histamine related symptoms yet still had DAO levels below the suggested normal values. Previously, levels below 10 U/mL were considered low. In this group, the median value was 11.3 U/mL. BMI and age were shown to have an impact on DAO activity with increasing activity of DAO mirrored by increasing BMI and age. The increase in DAO with increasing BMI was only seen among women. While no difference in median DAO was found between men and women, more women had DAO levels < 3 U/mL. No associations with the investigated comorbid conditions could be found in this group. 

The DAO enzyme has been described as a barrier against exogenous histamine. Clinical studies have published data on relations between low levels and histamine hypersensitivity. Levels that have earlier been considered as decreased levels of DAO were found in 44% of the current study group, suggesting that these earlier cut-off levels need to be reviewed. A recently published study of 146 patients with a diagnosis of HIT showed that the accumulated symptom severity was associated with low levels of DAO [10]. However, isolated skin or gastrointestinal symptoms were as common in the group with normal levels (>10 U/mL) as in the group with reduced levels.

In our study, we used a radio-extraction assay (REA) to evaluate the activity of DAO in a serum. However, there are also ELISA tests on the market that can measure the content of DAO in human biological products. ELISA methods have the advantage of being easy to implement, since the vast majority of clinical laboratories have experience with this type of assay. On the contrary, a REA method needs more experienced technicians and specialized instruments to read the results, and it is not widely available. 

Due to national regulations regarding use of radioactive tests in clinical laboratories, REA is difficult to obtain within the EU. Consequently, our laboratory also implemented an ELISA test and compared the results for the two methods. The correlation was good (Pearson R^2^ 0.87) but there were discrepancies between the two methods. One may speculate that this happens because, like many other enzymes, DAO can be affected by mutations that reduce its activity while preserving or not the epitopes that are recognized by monoclonal antibodies used in ELISA tests. However, if a lower DAO activity as a result of these mutations correlates with a clinical outcome in the form of histamine intolerance (HIT), it is still yet to be proven [16,17]. Hence, we can probably draw the conclusion that an ELISA test will detect most of the cases of DAO deficiency in order to support the diagnosis of HIT, but we cannot exclude for sure the possibility of losing some HIT cases if only ELISA is performed. Additional research in this field, where the clinical picture, REA, and ELISA tests are correlated, is of course warranted.

The current study has the advantage of being based on a large population-based cohort covering a wide age-range. It is also the largest study on distribution of DAO levels that has ever been published. However, it was not initially designed to investigate symptoms related to histamine-rich food. There is also a possibility of population bias that could potentially result in a higher prevalence of hypersensitivity and reported GI symptoms. This sample must also be considered regarding population bias since there might be a skewed representation of subjects with self-reported symptoms. However, a non-response study on the initial questionnaire revealed no bias regarding symptoms [18]. There was no difference between the groups available for DAO testing in this material (Table 1). The reported prevalence of food hypersensitivity in this population is similar, or only slightly higher, compared with other adult cohorts [19,20].

One out of five participants in the study declared use of antihistamines. H1-antagonists are often used in clinical practice to reduce histamine-related symptoms in the clinical setting. In a small study on healthy subjects, DAO activity did not seem to be influenced by H1-antagonists [21]. In this material, a stratified analysis including subjects not reporting use of antihistamines, no influence on DAO activity levels was seen.

### 4.1. The Natural History of DAO and Histamine Intolerance 

There is limited data available if DAO enzyme changes over a lifetime period apart from reports that DAO is produced by the placenta and elevated during pregnancy [22]. Estrogen hormones are known to interact with immune response including the release of histamine [23] and maybe there are other conditions that affect the response and affects the histamine metabolism. We found higher levels of DAO in older subjects but the since the group reflects all subjects, we could not correlate this finding to information regarding intolerance to histamine. It has been suggested that around 1% of the population suffers from histamine intolerance where 80% were middle aged [24]. However, this reference cannot be accessed in English, leaving further questions surrounding how the data was collected and if histamine intolerance is more common in people of older ages.

### 4.2. DAO in Relation to Allergy, Diet and Other Factors 

The impact of allergic disease on histamine intolerance and DAO activity has been elucidated in some smaller studies. Lower levels of DAO were found in a group of well-defined subjects with allergy compared to controls, but it was studied in a small group [25]. In a cohort study conducted by Cucca et al., food allergies and respiratory allergies were found to be more frequent in the group with DAO levels of >10 U/mL, which are suggested to be normal values [10]. It has also been suggested that a histamine-rich diet can affect other atopic conditions. In a pilot study, it was found that diet can have an active and direct impact on asthma symptoms in children [26]. In this cohort, no difference with regard to atopic conditions could be seen between patients with different DAO levels. It is notable that some of these factors affecting DAO on an individual level, both short- and long-term, may not be possible to assess on a group level in this type of cohort.

Some control studies described an increase in DAO when following a histamine-reduced diet, suggesting a recovery of enzyme levels [12,27]. If this is true, diet can be said to affect DAO levels. In a case-control study with oral provocation of histamine-rich meals, there was no observed difference in short-term activity of DAO between patients and controls, suggesting that there is no difference when metabolizing histamine [28]. In this present representative group of adults, we found no relation between levels of DAO activity in groups who reported a low intake of histamine-rich foods. However, less than 9% of the group, corresponding to 93 persons, reported any symptoms. Surprisingly, no difference in gastrointestinal symptoms could be seen between groups with low or normal levels of DAO. However, airway symptoms were more common among subjects with low levels (<3 U/mL).

In review articles, drugs including analgesic and anti-inflammatory agents are listed as interfering with DAO activity [4,5]. Furthermore, agents in anti-depressive drugs and medications used for hypertension, which are widely distributed among adults, have shown to have an inhibitory effect. We have information about usage of these medications in the study group, but the study is severely underpowered to draw any conclusions on a possible connection.

## 5. Conclusions

In the current population-based study, it was observed that nearly half of the participants exhibited a DAO level below the suggested normal levels. These findings suggest that these earlier cut-off levels need to be reviewed. Furthermore, BMI and age were shown to have an impact on DAO levels in a sex-dependent manner, suggesting the need for further studies to improve the use of DAO as a biomarker for histamine intolerance.

## Figures and Tables

**Figure 1 nutrients-15-02887-f001:**
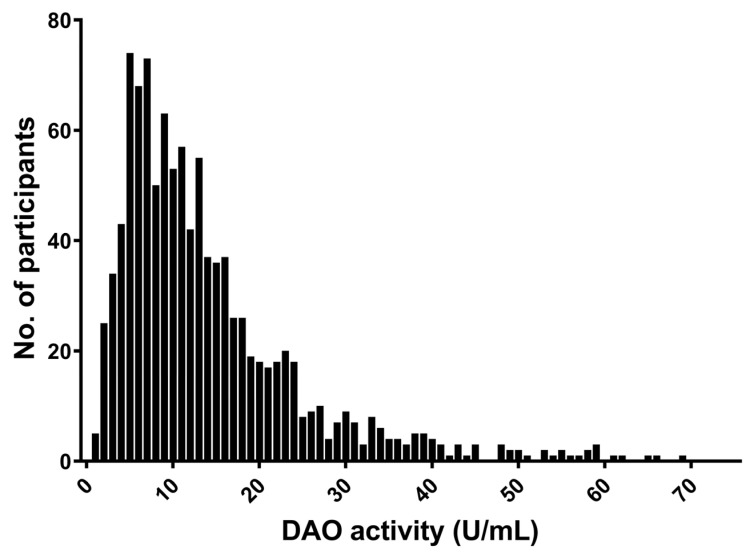
Distribution of DAO-activity in the study population. Five values above 60 has been excluded for clarity.

**Figure 2 nutrients-15-02887-f002:**
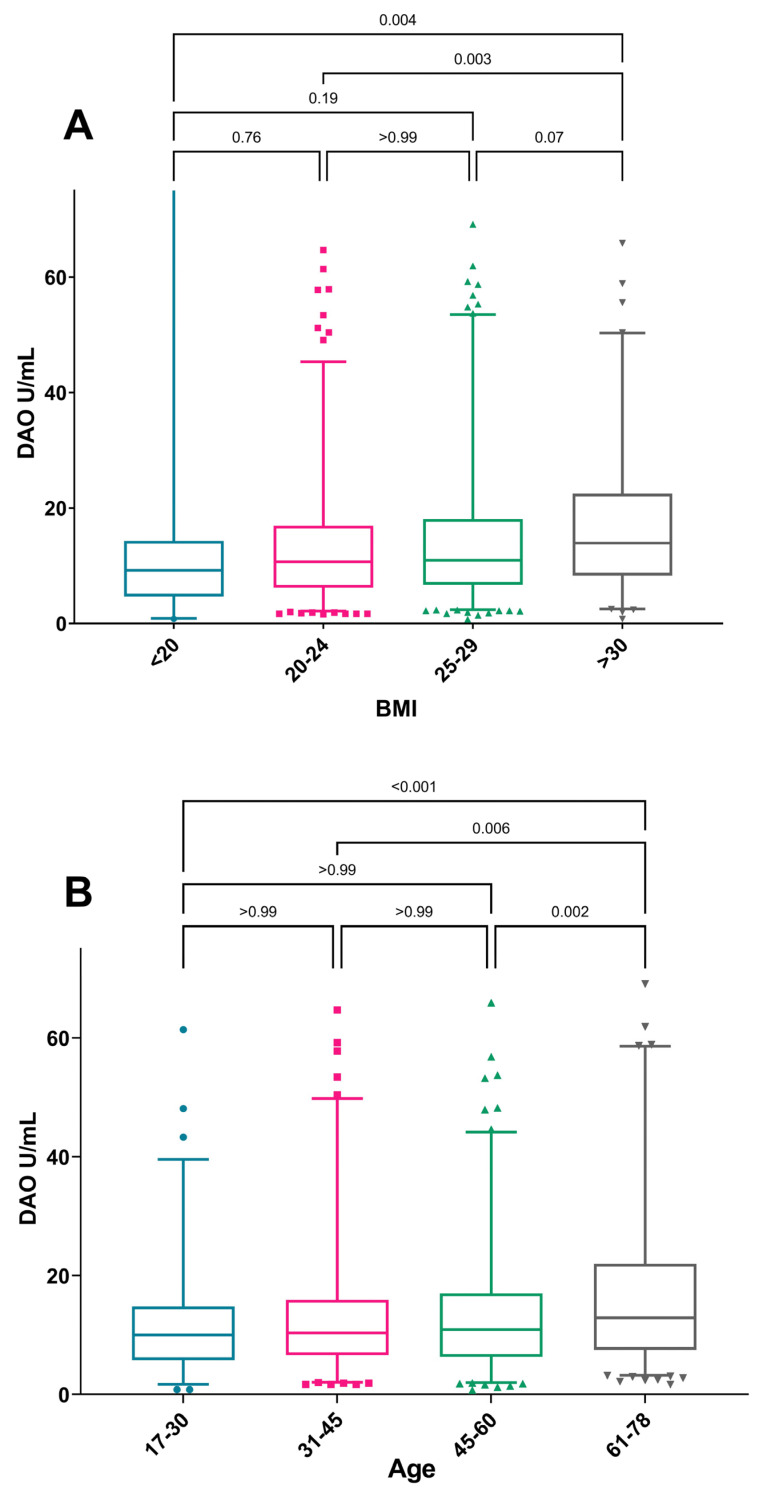
Median DAO-activity by (**A**) BMI category and (**B**) age category. The Mann–Whitney U-test was used for comparison of DAO median levels between age groups. Box-plot depicting range of Q1–Q3 and median with whiskers showing range of 2.5–97.5 percentile. Outliers shown as individual values.

**Figure 3 nutrients-15-02887-f003:**
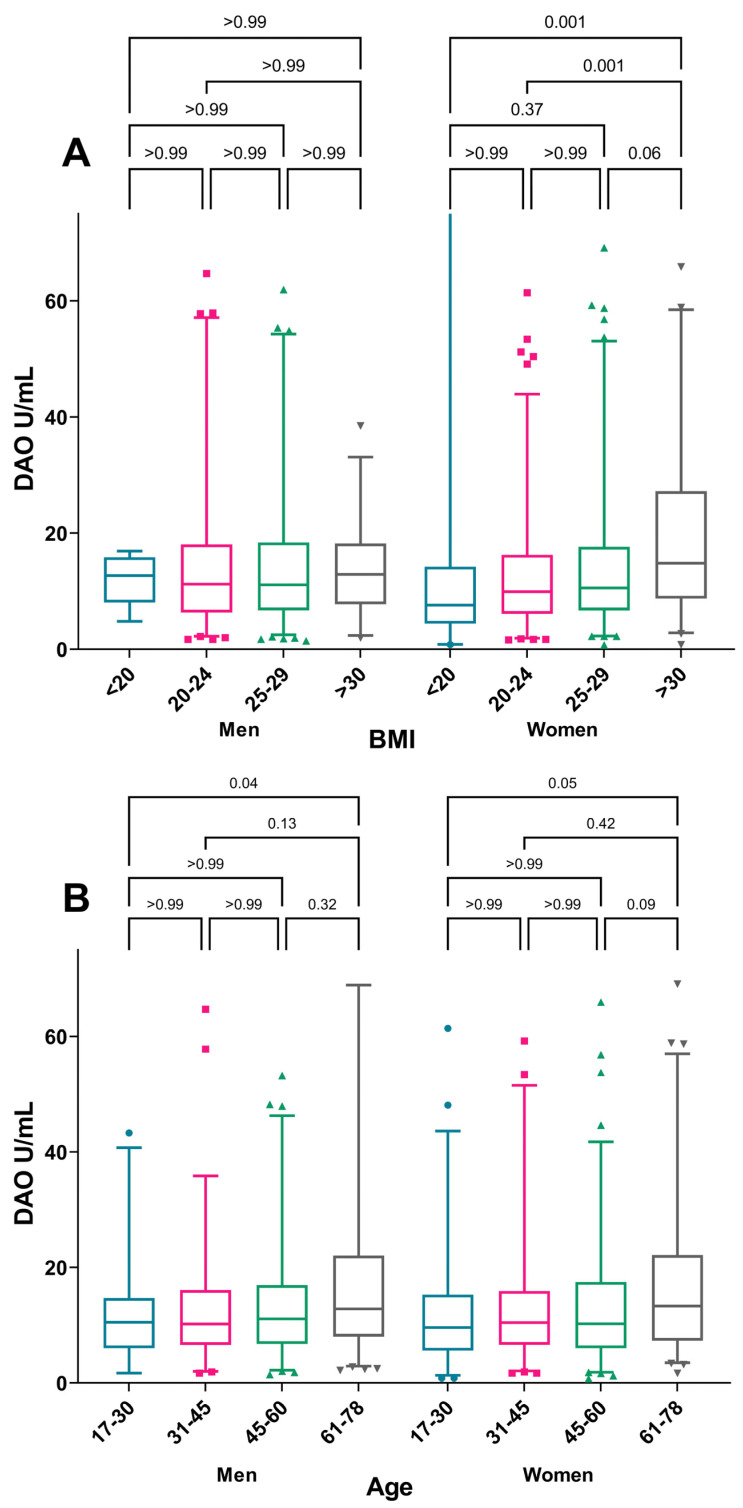
Median DAO-activity by sex, (**A**) BMI category, and (**B**) age category. The Mann–Whitney U-test was used for comparison of DAO median levels between age groups within each sex. Box-plot depicting range of Q1–Q3 and median with whiskers showing range of 2.5–97.5 percentile. Outliers shown as individual values.

**Table 1 nutrients-15-02887-t001:** Differences in demographic factors and comorbid conditions between subjects where DAO-activity was and was not analyzed.

Demographic Factors		DAO Analyzed N 1051	DAO Not Analyzed N 121	*p*-Value
Sex (%)	Females	52.9	61.2	0.101 ^
Age (mean ± SD)		50.7 ± 15.3	48.2 ± 15.6	0.100 ^
Age group (%)	17–30	14.7	18.2	0.099 #
	31–45	24.5	29.8	
	46–60	30.1	25.6	
	61–78	30.7	26.4	
BMI (%)	<20	4.5	4.1	0.103 #
	20–24	37.0	46.3	
	25–29	43.2	33.9	
	≥30	15.3	15.7	
Education level (%)	Primary school	14.9	9.9	0.132 #
	Secondary school	4.6	3.3	
	High school	34.2	40.5	
	University	46.1	45.5	
Comorbid diseases				
Asthma (%)		12.3	7.4	0.136
Sensitized to airborne allergens (%)		27.3	28.2	0.823
Sensitized to common foods (%) $		5.4	5.7	0.823
Used medication for hay fever (%)		27.8	31.4	0.395
Heart medication (%)		9.2	5.8	0.158
Hyperlipidemia (%)		10.0	8.3	0.630
Gastric reflux (%)		44.1	49.6	0.464

^ Student’s *t*-test to compare mean value between DAO analyzed/not analyzed. ^#^ Mantel–Haenszel test for trend to compare frequency between DAO analyzed/not analyzed. ^$^ Egg white, milk, fish, wheat, peanut, and soybean.

**Table 2 nutrients-15-02887-t002:** Demographic factors and comorbidities by DAO activity divided into groups.

			DAO (U/mL)		
		≥10	3–10	<3	*p*-Value
Total of included subjects (%)		55.7	39.8	4.6	
Sex (%)	Females	53.1	41.7	5.2	0.056 *
	Males	58.6	37.6	3.8	
Age group (%)	17–30	50.0	43.5	6.5	0.001 ^#^
	31–45	52.7	41.9	5.4	
	46–60	53.5	40.5	6.0	
	61–78	62.8	35.6	1.5	
Age (mean ±SD)		51.8 ± 15.3	49.6 ± 15.3	45.3 ± 14.7	0.003 ^
BMI (%)	<20	46.8	40.4	12.8	0.027 ^#^
	20–24	54.0	40.6	5.4	
	25–29	55.3	41.4	3.3	
	≥30	63.4	32.9	3.7	
Asthma (%)	Yes	54.3	38.8	7.0	0.430 *
	No	55.9	39.9	4.2	
Sensitized to airborne allergens (%)	Yes	54.8	38.5	6.7	0.296 *
	No	56.2	40.0	3.8	
Sensitized to common foods (%) ^$^	Yes	60.8	31.4	7.8	0.833 *
	No	55.2	40.7	4.0	
Used medication for hay fever (%)	Yes	51.5	42.3	6.2	0.041 *
	No	57.4	38.8	3.8	
Heart medication (%)	Yes	56.4	40.6	3.0	0.696 *
	No	55.7	39.5	4.8	
Medication for hyperlipidemia (%)	Yes	60.0	35.2	4.8	0.449 *
	No	55.2	40.2	4.6	
Gastric reflux (%)	Yes	56.6	39.7	3.7	0.448 *
	No	54.9	39.8	5.3	

* Fisher’s exact test to compare frequency between DAO-groups. ^ Student’s *t*-test to compare mean value in each DAO-group. ^#^ Mantel–Haenszel test for trend to compare frequency between DAO-groups. ^$^ Egg white, milk, fish, wheat, peanut, and soybean.

**Table 3 nutrients-15-02887-t003:** Logistic regression model for DAO including sex, age, and BMI as independent variables.

		DAO	
		<3	3–10
Sex (%)	Females	1.16 (0.62–2.17)	1.20 (0.92–1.55)
Age group (%)	31–45	3.84 (1.22–12.12)	1.49 (0.98–2.27)
	46–60	3.49 (1.21–10.08)	1.37 (0.97–1.95)
	61–76	4.30 (1.57–11.82)	1.30 (0.94–1.80)
BMI (%)	<20	2.30 (0.81–6.52)	1.01 (0.52–1.95)
	25–29	0.70 (0.34–1.42)	1.09 (0.82–1.46)
	≥30	0.69 (0.26–1.78)	0.74 (0.50–1.11)

Reference groups: aged 17–30, males and BMI 20–25.

**Table 4 nutrients-15-02887-t004:** DAO-activity (U/mL) among subjects reporting symptoms from histamine rich foods.

Symptoms from Histamine Rich Foods	Median DAO U/mL (IQR)	*p*-Value *
Gastrointestinal	11..4 (13.0)	0.6
Dermatological	14.3 (14.0)	0.197
Airway	9.6 (8.0)	0.215
Neurological	14.0 (7.5)	0.423
Any of above	11.9 (12.3)	0.215
None	11.2 (11.5)	n/a

* Mann–Whitney U-test was used for reporting symptoms from histamine rich foods compared with the group not reporting symptoms from histamine rich foods.

## Data Availability

The data that supports the findings of this study are available upon reasonable request from the corresponding author (JvO). The data are not publicly available.

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
