# Peer review of "The Use of DAO as a Marker for Histamine Intolerance: Measurements and Determinants in a Large Random Population-Based Survey"

_nutrients, 2023, doi:10.3390/nu15132887_

Round 1
Reviewer 1 Report
In this paper van Odijk and coauthors describe the distribution of the values of DAO enzymatic activity in a large representative cohort of adults to determine the association between this parameter and defined clinical phenotypes.
Choosing a large population of non-prescreened individuals in my opinion sets the stage for unbiased results
The method used for the measurement of serum DAO activity is the one supported by the larger amount of published evidence
The experimental design is sound, the statistical analysis is carried out properly, the discussion is comprehensive and the conclusions are convincing
I would suggest to indicate in the discussion a comment to a methodological point which is often overlooked. ELISA-based methods are available on the market, and are used with increasing frequency to generate publication on peer-reviewed Journals. Overall, the national regulation concerning the manipulation of tritiated thymidine (though at very low activity) is favouring the shift from radio-extraction methods to ELISA-based methods. There are presently no available studies to correlate measurements based on these 2 methodological approaches. This issue makes it even more important to implement further studies to improve the use of DAO as a biomarker for histamine intolerance in clinical practice.
Minor editing may be required. E.g., line 15: ...analysis of the activity of the enzyme diamine oxidase (DAO) is sometimes used.... This study is aimed to describe the distribution of the activity of DAO in a large...
Author Response
Many thanks for the comments regarding the methodology.
We have added a section in the discussion part regarding the use of ELISA- based methods compared to REA.
Reviewer 2 Report
There is still major confusion among physicians and patients about the diagnostic value of DAO measurements as a marker for histamine intolerance. Here, the authors present data on a standardized assessment of DAO activity in a large cohort of individuals. Thus, this work adds a valuable piece to the understanding and clarification of this widely used diagnostic test.
Minor comments:
Please specify from which "human samples" the analysis was done, e.g. serum?
In Table 1, it is not clear why the authors show characteristics of individuals for which DAO analysis was not available. It seems that this population was not assessed for any further analysis shown in the manuscript and, thus, could be omitted. Could the authors clarify the value of showing data about this subpopulation.
In Table 2, it is not exactly clear what the given p-values linked to. A more detailed explanation about the statistics and comparisions performed in this table is needed, e.g. in a table legend.
Check for some minor spelling errors.
Author Response
|
Many thanks for the comment regarding the use of this manuscript to elucidate further information about the test values. Data from subjects not having preformed the DAO analysis is used to illustrate the representativeness of the study sample from the original cohort and we think it provides valuable information about the study group. The minor comments have been corrected in the manuscript. The figure have also been provided as a separate file to enhance graphic quality. We also corrected the spelling errors and edited the language.
|
Reviewer 3 Report
Currently, the occurrence of undesirable symptoms after food consumption is one of the major health problems. The spectrum of abnormal reactions is wide, ranging from mild gastrointestinal symptoms, through skin symptoms of varying severity, to life-threatening anaphylaxis. Diagnostics is usually difficult and requires an individual approach to each patient. It is often necessary to differentiate symptoms dependent on immune mechanisms from food intolerances of a different etiology. In the case of some patients, even extensive diagnostics is not able to clearly establish the diagnosis. It is then suggested to consider histamine intolerance (HIT), also known as pseudo-allergy or diamine oxidase deficiency syndrome. It is worth emphasizing that in some patients HIT coexists with a typical allergic reaction.
The diagnosis of histamine intolerance (HIT) is usually established after exclusion of the causes of allergy in the presence of at least two clinical symptoms typical of allergies and their improvement/regression after a low-histamine diet. However, there are no clear diagnostic criteria or biomarkers for HIT. Diagnosis is made after exclusion of food intolerance, gastrointestinal diseases, IgE-mediated food allergies and mastocytosis. It is worth noting that some foods can be both a strong allergen and a rich source of histamine (e.g. fish and seafood, including shrimp). Sometimes the patient's symptoms have a mixed etiology, i.e. they are both the result of an IgE-mediated allergy and a disorder of histamine metabolism.
Some studies suggest that determination of serum DAO activity may be helpful in the diagnosis of HIT. There are publications showing that the concentration of DAO in patients with HIT is statistically significantly lower than in healthy people. Therefore, the measurement of DAO activity seems to be a good diagnostic strategy in the case of suspected HIT.
Measuring the level of histamine or DAO in the patient's blood is currently possible, but according to some authors it has significant diagnostic limitations. This is due to the fact that DAO activity may not differ in individual patients depending on their comorbidities and medications taken. DAO activity is also not constant in subsequent measurements and can vary significantly even in the same patient. Usually, these tests are not recommended in the early stages of the routine diagnosis of food intolerances. Measuring DAO activity may therefore only make it possible to establish the correct diagnosis in some patients. Genetic tests are also available for the two main pathways of histamine degradation in the body, including DAO and HNMT. However, there is a lack of high-quality research to support the effectiveness of these assays.
In conclusion, the possibility of using serum DAO activity as a marker in the diagnosis of HIT has been discussed for a long time. Research results and standardization of tests still raise many doubts. In my opinion, this article does not bring any new information to the literature on the subject. For this reason, I rate the originality of the reviewed manuscript as low.
Minor remarks:
Figure 1 - poor graphic quality
Table 2:
· hard to read,
· for Standard Deviation, the acronym SD is usually used,
· I suggest that the data be presented in the form of an mean +/- SD and statistical significance should be marked directly next to the data they refer to.
Table 3: I also propose to present the data in the form of an mean +/- SD. In decimal fraction notation, points should be used instead of commas.
P values can be omitted - just the information below the table about the lack of statistical significance.
Figure 2 and 3 - the caption lacks information on the statistical tests used for the analysis.
In addition, the manuscript requires editorial corrections.
Author Response
|
Thank you for a very elaborated summary of the topic. We do agree that this area still raise many doubts and that further research is essential. There is a considerable need to increase the knowledge of the subject among health personnel working within the area. This manuscript does not provide any causal evidence regarding the role of DAO within subjects with histamine intolerance but, to our belief, valuable information about the distribution of DAO in a large representative group of adults, never published before. In addition, the manuscript can also bring forward some correlations between DAO levels and factors like age and BMI that might be valuable when estimating the role of DAO in evaluations of patients. We do agree with lack of support that DAO activity may not differ in individual patients depending on their comorbidities and medications and that DAO may differ individually on short term. The current paper present data on group level and are not based on patients but rather individuals from a population sample. We have tried to clarify this in the manuscript. We do agree that other designs and study materials are needed to elaborate with the questions whether DAO tests can be useful in some patient groups. The minor remarks stated in figures and table texts like figure legends and acronyms have been corrected according to suggestions.
|
Round 2
Reviewer 3 Report
In the revised version of the manuscript, the authors addressed all my "minor remarks". In terms of editorial and presentation of the results, the manuscript was sufficiently refined to warrant publication in Nutrients.
However, I still believe that this article does not bring anything new to the literature on the subject.